# A Product/Service System Design Schema: Application to Big Data Analytics

**Tomohiko Sakao *** and **Abhijna Neramballi**

Division of Environmental Technology Management, Department of Management and Engineering, Linköping University, 581 83 Linköping, Sweden; abhijna.neramballi@liu.se
* Correspondence: tomohiko.sakao@liu.se; Tel.: +46-73-620-9472

**Abstract:** The challenge of environmental sustainability has required product/service systems (PSSs) to play a substantial role. New technologies such as big data analytics (BDA), which have high potential to improve or enable PSSs, are increasingly implemented in industry. However, research achieved in the past and research opportunities in the intersection of PSS design and BDA are unclear in the literature. Therefore, this article took an inter-disciplinary approach and aimed to pave the way forward for research and development in PSS design and show opportunities to improve PSS design and delivery using BDA. The research methods adopted were literature synthesis and systematic literature review. The synthesis of PSS design literature resulted in a schema consisting of 10 design steps for PSS conceptual design. The systematic review of BDA literature found 11 research works, including industrial applications, which were then mapped on to the PSS design schema. This revealed the achievement of applied research using BDA for some of the PSS design steps as well as opportunities of research for the others. The two inter-related areas of research, PSS design and BDA, were connected with each other more clearly, so that further research could be anchored and motivated with more specificity.

**Keywords:** systematic literature review; literature synthesis; customization; environmental sustainability

## 1. Introduction

Product/service system (PSS) design has been heavily researched in the last two decades, as reviewed by (e.g., [1–3]), and many concepts [4,5], models [6,7], methods [8,9], and tools [10,11] for PSS design have been proposed in the literature. One of the origins of PSS design research in academia is environmental sustainability (e.g., [12]), and the environmental implications of PSSs have long been debated [12,13]. We acknowledge the advancement of scientific insights and industrial practice [14], and some of the academic knowledge has been used effectively in industry. Even effective implementation of a PSS design method in a prescriptive manner in an international manufacturing company has been reported [15].

Today, thanks to industry's increasing interest in big data (big data can be recorded), IoT (Internet of Things; things can be connected, and thereby the big data can be collected efficiently), and big data analytics (the collected data can be analyzed to produce useful information; BDA, hereafter) as reviewed by, e.g., [16], we see an enormous opportunity to further advance the practice of PSS design and delivery in industry. This opportunity is evident in industrial reports: e.g., a report in the UK [17] listed these technologies of importance to further enhance PSS, while another in Germany [18] expects different types of services enabled by various technologies of Industry 4.0. Now, a question arises as to how technologies such as BDA could improve PSS design and a resulting delivery: Does such a PSS design process already exist that can effectively accommodate the outputs of BDA? Is there a need to amend existing PSS design processes to exploit the obtained information? Further, it is of high

interest to ask: What are the environmental implications of applying BDA in PSS design? At present, no clear answer to these questions can be found in the literature, although some insights on how to sense, record, access, and utilize such data in servitization (i.e., a process for a firm to increase service provision [19]) have been more available (e.g., [20–22]).

This article aims to pave the way forward for research and development in PSS design and show opportunities to improve PSS design and delivery using BDA. The research methods adopted are literature synthesis and systematic literature review (to be detailed in Section 3). Literature synthesis is a process aiming to present insights in a form that can be effectively exploited by users in other disciplines or practitioners but requires more scientific development (as will be explained in Section 2.3). The literature used is comprised of scientific journal articles. First, insights into PSS design are consolidated as a generic PSS conceptual design schema through an intra-disciplinary literature synthesis. Subsequently, past achievement and future development opportunities for applying BDA to improve PSS design and delivery are identified in the PSS conceptual design procedure by systematic literature review and inter-disciplinary literature synthesis. The guiding research questions (RQs) are formulated as follows:

1. What relationships exist between PSS design and BDA?
2. What are the achievements thus far and development opportunities in PSS design through applying BDA?
3. What does the literature report on the environmental implications of applying BDA in PSS design?

The contributions of this article are threefold. First, a consolidated PSS design procedure that is described in generic terms yet maintains the essence and thereby has high applicability is provided (in Section 4.1). It means that this procedural schema will be applicable not only to various types of PSS design cases but also to exploitation in disciplines other than PSS design. Second, this article gives one perspective nexus between PSS design and BDA in an inter-disciplinary research approach [23], as well as presents an account of review for PSS design enhanced by BDA (in Section 4.2). Third, the outcomes of inter- and intra-literature syntheses presented as a concrete case are demonstrated, contributing to the further enhancement of literature synthesis as a scientific research method.

## 2. Theoretical Background

### 2.1. PSS Design

A PSS can be defined as a mixture of tangible products and intangible services that are designed and combined so that they are jointly capable of fulfilling specific customer needs [4]. The efficacy of a potential PSS in terms of, e.g., its environmental performance and customer value, is generally determined in the conceptual design phase. This is shown by the current body of knowledge, arguing that the conceptual design phase is considered to be critical for the development of PSSs (e.g., [24]). During PSS conceptual design, characteristically distinct product and service elements are expected to enter the design space [25], and these elements are then prescribed to be integrated systemically [26]. Therefore, PSS design requires concurrent and integrated approaches to facilitate the tight coupling of the constituent elements [27]. Consequently, PSS design is supposed to require dedicated and characteristically distinct approaches, in comparison to product design (e.g., [28–30]). Note that concurrent design [31] and integrated product development [32] differ from PSS design, as they lack the simultaneous and integrated design of products and services (see more explanation in [33]).

Review articles have been published describing extensively extant prescriptive PSS design support, methods, approaches and tools (e.g., [1,2,26,34]). Further, a few other publications have reviewed, analyzed, and synthesized the commonalities recurring in the state of the art in prescriptive PSS design research. For example, Cavalieri and Pezzotta identified and categorized the various aspects of PSS engineering into the following three fundamental elements: entities, lifecycle, and actors [35]. Clayton, Backhouse, and Dani analyzed existing approaches for overall PSS design and synthesized

the common phases within these approaches [27]. However, none of the previous works synthesized and consolidated the vastly disparate qualitative dimensions of the prescriptive PSS design literature in a generic form that can be used by practitioners to effectively guide their conceptual design activities. In order to realize the impacts of qualitative research, it is essential to situate the gathered insights in a larger interpretive context and to present it in an accessible and usable form in the real world of practice [36].

### 2.2. BDA (Big Data Analytics)

Big data was defined with high volume, velocity, and variety by Laney [37]. Later, the definition was developed to also include veracity, value, and variability [38]. BDA (big data analytics) can be defined as a holistic process that involves the collection, analysis, use, and interpretation of data for various functional divisions with a view to gaining actionable insights, creating business value, and establishing competitive advantage [39]. In relation to manufacturing in a broad sense, a variety of subjects are relevant to BDA, as reported by a recent review [16], and all the lifecycle phases, in principle, have the potential to benefit from BDA, such as marketing, design, production, logistics, use, maintenance, and end-of-use treatment. The manufacturing industry has increased interest in BDA, and earlier literature reported industry-relevant applications in different areas, such as factory operations (e.g., [40]) and operations management (e.g., [41]). However, using BDA effectively in the decision-making process remains a challenge in general [42]. Concerning PSS design in particular, little literature has reported BDA applications, and few articles have presented their possibility (e.g., [43,44]).

BDA can be effective for PSSs, because PSSs are characterized by the involved uncertainty in the use phase [45] and this uncertainty could be decreased substantially by BDA. This potential effect could be represented by PSS customization (e.g., [46,47]): BDA empowers frontline employees to customize services and is re-enabling the companies to balance standardization versus customization of services, as explained in [48]. The customization (individualization) of products, services, and PSSs is indeed reported as one of the benefits of Industry 4.0 technologies, including BDA based on interviews with managers in industry [49].

### 2.3. Systematic Literature Review and Literature Synthesis

For systematic literature review as a research method, several seminal works are available. For instance, a standard process with three stages for systematic literature reviews in management research is documented [50]. Systematic literature reviews have been practiced for many years in different academic disciplines. A potential issue has been pointed out, though: "authors of literature reviews are at risk for producing mind-numbing lists of citations and findings that resemble a phone book—impressive case, lots of numbers, but not much plot" [51]. To address this issue, making a conceptual model used throughout a review article may be effective, as emphasized in the field of information systems [52]. In addition, the systematic literature review process is supposed to include synthesis [50], and its aim is articulated to make impacts by being presented in an accessible and usable form in the real world of practice and policy-making [36]. Building upon a model and making impacts are highly sensible for a review article, as an author of a good review article is more than simply a reporter [53].

Despite these pedagogic works of literature published around the last two decades, assessment of the impacts of review articles remains largely masked. An issue with the general quality of review articles, in general, is not new [54]. However, this issue deserves more attention now, as we face an "insanely" growing number of review articles (e.g., as reported in [55] (annual publications for systematic reviews in PubMed between 1991 and 2014 increased by 28 times, and many of them were concluded not to be useful)), but we are expected to conduct academic research for societal challenges in the real world [56] (often called trans-disciplinary research [23]): the recipients of the insights in an academic article are not only academic researchers in the concerned discipline, but also researchers in other disciplines and, more importantly, practitioners in industry and governments. For instance,

in health research, a need was recently found to synthesize many *review* articles in a manner that is accessible and usable to practice [57]. Systematic literature reviews, to tackle the urgent societal demand of environmental sustainability, need to increase their impact, especially by improving the synthesis part.

## 3. Research Method

### 3.1. Overview

The research procedure adopted in this article is depicted in Figure 1. It consisted of two steps, i.e., intra-disciplinary literature meta-synthesis in PSS design and the systematic literature review and inter-disciplinary meta-synthesis of BDA and PSS design literature. Each of them is explained in Sections 3.2 and 3.3, with the results of the two steps shown in Sections 4.1 and 4.2. The first step is a concrete example of intra-disciplinary literature synthesis and consolidates a generic procedure, i.e., a schema for conceptual PSS design. This schema was used as a model for the systematic literature review and inter-disciplinary meta-synthesis of BDA and PSS design literature.

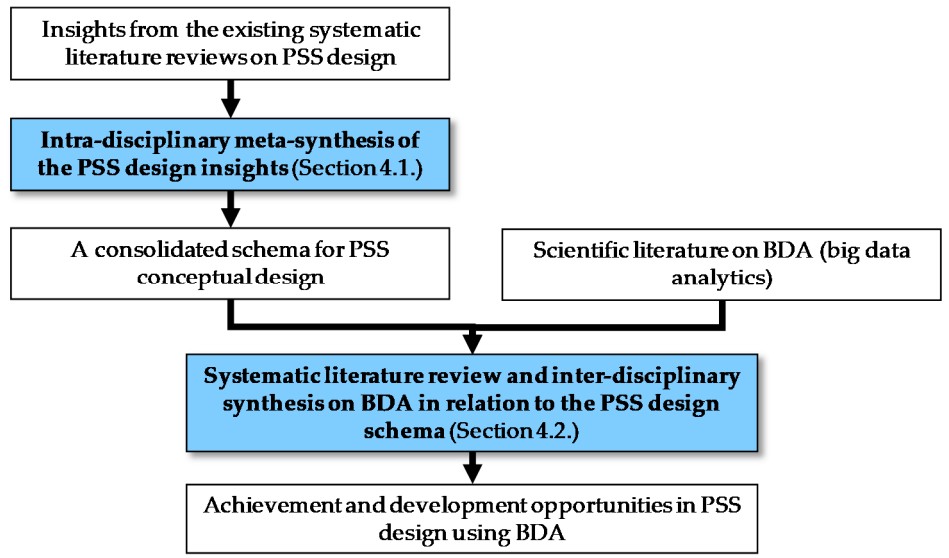

**Figure 1.** Research procedure.

### 3.2. Intra-Disciplinary Meta-Synthesis and Consolidation of Prescriptive PSS Design Insights

Since there are several past reviews of prescriptive PSS design literature, a meta-analysis and a subsequent meta-synthesis of these reviews were carried out to extract insights from the available knowledge. Meta-synthesis is useful to extract narratives, theories, ideas, interpretations, or generalizations from a set of qualitative studies that are centered around a specific topic of interest [36]. This type of synthesis can promote the consolidation of the knowledge available in the state of the art, which can serve as a basis to guide both industrial practice and future research.

The existing reviews of PSS design literature were initially identified in a database search with keywords that were centered around the PSS design topic. The ISI (Institute for Scientific Information) Web of Science was the scientific database chosen, since it provides journals with impact factors and has been used with a similar purpose in a number of other reviews. The scope for the search focused on publications in the Web of Science Core Collection, where only the databases SCI (Science Citation Index) Expanded and SSCI (Social Sciences Citation Index) were selected, as these were relevant to the topics of interest. The years selected were between the earliest possible in the database, 1975, and the most recent, 2020. The list of chosen keywords and the respective combinations was as follows: (("Product-service system*" OR "PSS*" OR "Integrated product service system*" OR "functional products*" or "Integration of product and service design*" or "Integrated product service

offering") AND "design*" AND ("literature review*" OR "review*")). This search was refined by excluding several unrelated Web of Science categories (see Appendix A), which yielded 141 results. The search was further refined by filtering the type of publication as *review article*. The refinement yielded 41 results. These hits were further narrowed down to 10 final literature reviews by applying the following inclusion criteria: the selected publication should include an extensive review, analysis, and subsequent description of existing prescriptive PSS design support methods, tools, or principles. Subsequently, the selected reviews were subjected to meta-analysis and synthesis to identify the commonly recurring facets of the extant prescriptive PSS design literature. This type of meta-synthesis is termed "intra-disciplinary meta-synthesis".

The collected insights from the synthesis of prescriptive PSS design literature were consolidated in the form of a prescriptive PSS design schema. The prescriptive design schema refers to a structured outline of procedural aspects to be considered during conceptual PSS designing that are relevant to the PSS design domain. This prescriptive schema is expected to support the designers with PSS domain-specific knowledge and thus to enhance their cognitive processes during conceptual PSS designing. Such a type of domain-specific prescription is deemed necessary to support practitioners who are not experienced in PSS design, as problem-solvers with limited experience in a specific problem area are claimed to require support specific to that problem domain, rather than generic problem-solving support strategies [58]. Furthermore, since this schema synthesizes and consolidates the essence of vastly disparate prescriptive design support, it could also be potentially useful towards a unified ontology for developing comprehensive computational PSS design knowledge support systems. Such a unified schema was deemed necessary to advance PSS design research by a review [34]. This was accomplished by pouring the extracted insights of prescriptive PSS design into the mold of a widely-used systematic approach to designing by Pahl and Beitz (PBSA: Pahl's and Beitz' systematic approach) [59]. More specifically, the conceptual design phase of PBSA was utilized as a framework to pour in the insights from PSS design. The PBSA outlines the following sequence of steps to be carried out during the conceptual design:

1.　Abstract to identify the essential problems;
2.　Establish function structures: overall function—subfunctions;
3.　Search for working principles that fulfil the sub-functions;
4.　Combine working principles into working structures;
5.　Select suitable combinations;
6.　Firm up into principle solution variants;
7.　Evaluate variants against technical and economic criteria.

Although PBSA has been applied mainly to products, its outline is expected to be able to serve as a mold for PSS design with the following reasoning: function includes teleological representations that can cover any expression related to potential purposes of the artefact [60], and based on this, Kannengiesser and Gero analyzed how the function issue is addressed in engineering, software, and service design each (ibid). Eisenbart, Gericke, and Blessing [61,62] also analyzed how the notion of function is applicable to other design objects than products such as software, services, and PSSs [60–62]. Other aspects, such as finding the working principles for the function, are also understood as applicable to other domains.

*3.3. Systematic Literature Review and Inter-Disciplinary Meta Synthesis of BDA Literature with the PSS Design Schema*

A systematic literature review, as documented in [50], was followed as a method. The literature on BDA to be analyzed was identified by a database search in January 2020 with keywords that were relevant to the subject and commonly used. The details of the database search of this systematic review were the same as the one used in Section 3.2. The keywords adopted were ("big data analytics") AND ("product" OR "service" OR "product service system") AND ("design" OR "development"): these

were applied for the *topic*. Out of the hits from the search, articles were identified after their titles and abstracts as well as their accessibility to get the set of articles relevant to BDA and design.

Subsequently, an inter-disciplinary meta-synthesis of BDA literature in relation to PSS design schema was carried out. Each article in this set was then assessed in terms of its contribution to the conceptual design of products, services, or PSSs, as well as its application to an industrial case. Here, the evidence from industrial cases was considered, because the applicability and effects of BDA depend highly on contexts. This criterion originated from the target recipients, including practitioners in industry in the spirit of trans-disciplinary research series [56]. In addition, the aspects of contribution to design by each article were documented: quality, cost, time, and environment. The main output was a collection of major findings grounded on industrial cases with their aspects of relevance.

With the authors' own analysis of this output, future development opportunities for PSS design were derived as an outcome of the inter-disciplinary literature synthesis. The main output is presented in the form of a matrix composed of the steps of the PSS design procedure and the aspects of the contributions in the past.

## 4. Results

### 4.1. Intra-Disciplinary Meta-Synthesis of the PSS Design Insights

#### 4.1.1. Meta-Analysis and Synthesis

A total of 10 reviews [1,2,26,34,35,63–67] were found relevant for capturing prescriptive PSS design insights out of the 41 search results that fit the defined inclusion criteria in Section 3.2. These 10 reviews described existing publications in the state of the art which presented prescriptive PSS design support (including methods, tools, and principles). The commonly recurring aspects peculiar to PSS design, which were supported by the reviewed publications within the 10 reviews, were found to include: 1. functionality-oriented designing, 2. identification of relevant actors along the lifecycle of PSSs, 3. value propositions, 4. development and integration of system elements, and 5. examination of the balance of the integration. Although these commonly recurring facets do not represent an exhaustive list of aspects relevant to PSS design, they potentially represent its distilled essence, as they are frequently addressed in the state of the art. Meta-analysis of the occurrence of these five aspects across the 10 reviews of the state of the art is documented in a concept matrix [52] in Table 1. The scheme for this meta-analysis was as follows: the review being analyzed should have shown one or more publications presenting a PSS design support that addresses the facet of PSS design.

**Table 1.** Concept matrix for the five facets of product/service system design recurring in reviews.

| Facet of PSS design | [1] | [2] | [26] | [34] | [35] | [63] | [64] | [65] | [66] | [67] |
|---|---|---|---|---|---|---|---|---|---|---|
| 1. Functionality-oriented designing | ✔ | ✔ | ✔ | ✔ | ✔ | ✔ | ✔ | ✔ | ✔ | ✔ |
| 2. Identification of relevant actors along PSS lifecycle | ✔ | ✔ | ✔ | ✔ | ✔ | ✔ | ✔ | ✔ | | ✔ |
| 3. Value propositions | ✔ | ✔ | ✔ | ✔ | ✔ | ✔ | ✔ | ✔ | ✔ | ✔ |
| 4. Development and integration of system elements | ✔ | ✔ | ✔ | ✔ | ✔ | ✔ | ✔ | ✔ | ✔ | ✔ |
| 5. Examination of the balance of the integration | ✔ | ✔ | ✔ | ✔ | | ✔ | ✔ | | ✔ | ✔ |

Note: The symbol ✔ means the facet was addressed by the reference.

These five facets are explained as follows. First and foremost, PSS development generally focuses on the delivery of functionality or satisfaction that the customers or users expect, instead of focusing on any specific product that can fulfil this functionality [68–70]. Thus, the identification and definition of functionality of the system being designed are considered to be crucial [69,70] in the initial stages of PSS design. This feature of PSS design is a fundamental enabler towards a design solution with dematerialization and thereby enhanced environmental sustainability (common with functional economy [12]). Second, a PSS is regarded as a socio-technical system [71,72] in which the expected results or functionality catalyzes the participation of various actors [73]. Consequently, identification of these relevant actors is essential to PSS designing [73], which is often addressed as part of a business model. Tight integration of actors such as manufacturers, service providers, suppliers, customers or

users, and other relevant societal actors identified along the lifecycle of the potential PSS is essential to PSS designing [10,74].

Third, value proposition based on identified use scenarios and requirements of the actors [73,75,76] is often recommended to be included. Value identification and proposition is considered as one of the crucial dimensions of PSS design across the literature. Value or costs represent a subjective interface between the involved actors in terms of positive or negative changes introduced by the functional unit of the PSS to the customer [77] and/or other relevant actors. This is also a critical part of a business model.

Fourth, potential realization structures need to be tightly integrated by the designers from a systems and lifecycle perspective [8,78,79]: the realization structures constitute the system elements, which include tangible products and intangible services, delivery channels, and infrastructure [74]. Fifth, in relation to the aforementioned integration, the balance of the integration of the system contents needs to be examined with regard to aspects such as value, costs, and quality [1,8,77,80]. This examination is expected to lead to optimization of the integration through exchanging efforts between products and services [81]. These five aspects are prescribed by the state of the art to be considered throughout the different stages of PSS development. Consequently, it is crucial to incorporate the identified facets in the conceptual design phase, the outcome of which will serve as a blueprint for the effective development of the PSS.

As found in the meta-analysis of the reviews to date aiming to show the essence of PSS design, the recurring aspects originate largely in the pre-2010 literature. This plateau in finding the essences may be seen associated with the research regarding environmental contributions of PSSs; the research published after 2006 mainly used case study research and simply seems to confirm the findings of the pre-2006 literature [67]. Obviously, even during the plateau, the essences for PSS design have been exploited, especially in applied research, to change industrial practice (e.g., references [15,82] could be regarded as best practice examples) and to address societal needs, such as the transition to a circular economy [83], and thereby the entire body of knowledge has been improved. In addition, advanced information and communication technologies, including IoT, BDA, and cyber-physical systems, have brought real-life feasibility to some of the visionary features of PSS design or increased efficiency in realizing the features. This reflection supports the research procedure adopted for this article (depicted by Figure 1).

4.1.2. Consolidation of Synthesized PSS Design Insights

The results of the meta-analysis were consolidated in the form of a prescriptive PSS design schema. This schema was developed by pouring the distilled essence of PSS design from the meta-analysis into the mold of PBSA. The PSS design schema represents a practical guide in terms of aspects to consider in a structured manner during conceptual PSS designing. It is represented as an iterative sequence of 10 steps and is expected to be applied during conceptual PSS designing in a methodical yet flexible manner. Some of the 10 steps are peculiar to conceptual PSS designing in contrast to conceptual designing by PBSA. This consolidation of the schema and relationships to PBSA conceptual design are represented in Table 2. For instance, the essence, 5. Examination of the balance of the integration is most applicable to Steps 5 and 6 of PBSA, and after the application Steps 7 and 8 of the PSS design schema were obtained.

Each of the ten steps of the PSS design schema is explained as follows. Step 1: Functional unit definition—designers are initially prescribed to abstract the problem they are addressing on a higher level and to identify functions and sub-functions that can address this problem. This step was added because of the importance of the first essence from the environmental perspective, i.e., functionality-oriented designing. Based on this, they are prescribed to assign a functional unit to guide their design process. A functional unit was originally defined as a measure of the performance of the functional outputs of the product system [84], which was extended to the PSS being designed in this context. It represents the measure of the teleological aspects of the PSS being designed in terms of the

expected performance or results, rather than the specific contents of the PSS. It provides a reference to which the inputs and outputs of the PSS can be related. By assigning such a unit of reference in the early stages, it allows the designer to increase the range of potential values (see Step 4) and design solutions that can fulfil the set function. During new design, identification of such a functional unit might be challenging, and designers might need to iterate when needed. Assigning such a unit also allows designers to compare the overall environmental impacts of different PSS concepts throughout their lifecycle, per unit of functionality to be delivered. It further helps designers to effectively address the issues of functionality and lifecycle during PSS designing. An example of a functional unit for a PSS can include "access to 20 km long mobility per day".

**Table 2.** Representation of the consolidation of product/service system (PSS) design schema.

| PBSA Framework for Conceptual Design | Distilled Essence of PSS Design | Consolidated Schema for Conceptual PSS Design |
|---|---|---|
| 1. Abstract to identify the essential problems | 1. Functionality-oriented designing | 1. Functional unit definition |
| 2. Establish function structures: overall function—subfunctions | 2. Identification of relevant actors along the PSS lifecycle | 2. Stakeholder identification<br>3. Requirement consolidation |
| 3. Search for working principles that fulfil the sub-functions | 3. Value proposition | 4. Value proposition |
| 4. Combine working principles into working structures | 4. Development and integration of system elements | 5. Criterion identification<br>6. Element integration |
| 5. Select suitable combinations | 5. Examination of the balance of the integration | 7. Balance examination |
| 6. Firm up into principle solution variants | | 8. Selecting combinations |
| 7. Evaluate variants against technical and economic criteria | No essence applicable | 9. Evaluating combinations<br>10. Solution selection |

Step 2: Stakeholder or actor identification—in the second step, designers are prescribed to identify all the relevant stakeholders or actors from a systems perspective who are involved in the value chain of the offering being designed. Examples of actors can include customers, suppliers, the environment, or users.

Step 3: Requirement consolidation—in this step, designers are prescribed to identify the use scenarios and consolidate potential requirements of the identified stakeholders or actors per unit of functionality for the PSS concepts being designed. Examples of requirements for the environment can include reduced emissions (these could be obtained from life cycle assessment results of an existing offering) and for customers' timely access to mobility. Requirement identification is a necessary step for the value proposition in PSS designing.

Step 4: Value proposition—after identifying the requirements, designers are expected to identify potential value propositions. Value in this context is considered as the trade-off between various perceived benefits and sacrifices or costs [85]. Examples of value for a customer can include the trade-off between a benefit, such as "pay only per use of functionality", and a cost, such as "no ownership of the product". The value proposition is also considered as a source of market differentiation for PSSs. It is a crucial dimension of the business model. The potential values to be generated need to be defined in the early stages of PSS design for effective delivery, since it will often be reliant on the design of the contents of the PSS.

Step 5: Criterion identification—in this step, relevant criteria for evaluating potential PSS concepts are established based on the identified requirements and value. Examples can include waiting time for users, carbon footprint, or cost per use. This step will contribute to clarifying the proposed values and gives the criteria the potential solutions will be evaluated against later in the design.

Step 6: Element integration—in this step, designers are recommended to conceptualize and integrate, from a systems and lifecycle perspective, the contents of the PSS that, in combination, can effectively address the requirements and facilitate values to the recipient actors. This integration also influences the business model of the PSS to be delivered (similarly to [86]). The contents can include products, services, and the channels that facilitate them. These contents are characteristically

heterogenous. Examples of element integration can include "designing a product for disassemble-ability and providing skilled service technicians, to facilitate faster repair".

Step 7: Balance examination—the balance of the integration of the elements is examined from a systems and lifecycle perspective. Since PSS is a system with interdependent product and service elements, it opens up the possibility to balance the integration. For instance, the expected lifetime of a specific part and the interval of replacing the part may be together examined for better balance.

Step 8: Selecting combinations—by now, designers are expected to have generated several alternative combinations of the system contents that address the requirements and increase value to the involved actors. In this step, the designers are prescribed to select specific combinations based on the balance examination.

Step 9: Evaluating combinations—in this step, designers evaluate the selected combinations of system contents against the criterion established in Step 5.

Step 10: Solution or concept selection—in the final step, designers select multiple combinations of the system contents that together address the requirements and value propositions, while fulfilling the identified functional unit.

### 4.1.3. PSS Design Benefiting from BDA

This section theoretically categorizes the data sources of BDA in relation to their applicability to the PSS design object. The design objects are either products or services. Although the data can be used to support the integration of the products and services, the major application of the data and resulting information will be largely focused on products, services or both. The sources of big data originate mainly from 1) the use of products and 2) the production and use of services. The first source of big data is dependent on product behaviour during the use phase, which is characterized by physical deterioration of the product over time as a consequence of several factors, such as use environments and user behaviours. Thereby, big data on the behavior of different products can be collected. The second concerns services. Services are inherently heterogeneous [87] and therefore have the potential to produce big data. Also, services are inseparable (ibid) and, therefore, the production and use form one source. These lead to four major types of BDA applications, as shown in Table 3. Note that the big data obtained from the production of products has the potential to be used for designing products or services. For instance, analyzing the deviation that is inevitable in the processes of transforming the design information into physical objects could yield adjustment of material composition decided in the detailed product design. Also, the degree of deviation of an individual product within the given tolerance may be used for adjusting the MTBF (mean time between failures) of the product decided in the detailed service design. However, focusing on the *conceptual* design of *PSS*, the relevance seems to be marginal. Other sources from the lifecycle, such as logistics, are also possible, but this article focuses on major types of BDA applications for PSS conceptual design.

**Table 3.** Major types of how PSS conceptual design benefits from big data analytics (BDA).

| Type | Data Source | Design Object |
|------|-------------|---------------|
| PP | Use of **p**roducts | **P**roduct |
| PS | Use of **p**roducts | **S**ervice |
| SP | Production and use of **s**ervices | **P**roduct |
| SS | Production and use of **s**ervices | **S**ervice |

Combining the four types with the 10 steps may create a variety of examples of how BDA results can be applied to each step of the PSS design schema. Before linking the output of the literature synthesis to that of the systematic BDA literature review, simplifying the output will increase the efficiency of the later step. Therefore, the potential for the four types of application of BDA to support the 10 steps was qualitatively assessed: Table 4 shows which combinations are relevant.

**Table 4.** Theoretical relevance of each step of the PSS design schema.

| | PP (Product to Product) and PS (Product to Service) | SP (Service to Product) and SS (Service to Service) |
|---|---|---|
| Step 1: Functional unit definition | T | T |
| Step 2: Stakeholder identification | | T |
| Step 3: Requirement consolidation | T | T |
| Step 4: Value proposition | T | T |
| Step 5: Criterion identification | | |
| Step 6: Element integration | T | T |
| Step 7: Balance examination | T | T |
| Step 8: Selecting combinations | | |
| Step 9: Evaluating combinations | T | T |
| Step 10: Solution selection | | |

Note: See Table 3 for the full meanings of PP, PS, SP and SS. T means theoretical relevance.

BDA application to the data obtained from the use of products (PP and PS) as well as that from services (SP and SS) is expected to have the potential to support Step 1 of the PSS conceptual design schema. For the former, a functional unit to be targeted for a PSS being designed can be defined, e.g., based on the BDA of the mean time to failure (MTTF) of a large number of industrial machines with more rationale. For the latter, e.g., BDA with the data of mobility service can potentially inform of the transport distance to be targeted quantitatively. This way, the functional unit could be defined more rationally.

Service data (SP and SS) are expected to be useful for stakeholder identification (Step 2). For example, BDA sourced from customer services can provide insights into the demographics of the customers and their usage patterns, which will support the identification of relevant actors and their characteristics. The data from products in use, as such, are not deemed highly relevant for stakeholder identification, because the data do not necessarily concern the users or stakeholders.

For the requirement identification and value proposition (Steps 3 and 4), BDA application based on products and services (types PP, PS, SP, and SS) is expected to have the potential of support. For example, BDA to identify trends in customer feedback or reviews for services (SP and SS) that include what is (dis)liked in what circumstances can potentially support the two steps. Also, the BDA of product performance in terms of breakdown or efficiency of products in use could be useful to determine the characteristics of products and services (PP and PS). Further, big data regarding customer use patterns (PP and PS) could be used to propose value-adding pricing offers or additional services for specific actors.

Step 6 (element integration) includes the conceptualization of products and services. Especially in conceptualization, the data from products and services are considered to be useful (types PP, PS, SP, and SS). For instance, BDA on product deterioration on a circumstance in question can be effectively useful for determining concepts for product structures. Further, based on this information, relevant services that can potentially reduce product deterioration can be conceptualized and integrated with the product structures.

To examine the balance of the integration of product and service elements (Step 7), the data from both products and services are potentially useful (types PP, PS, SP, and SS). For example, when the product malfunction probability becomes marginal (often faced in the case of new product introduction) in certain circumstances, locating service staffs close to the user sites on a constant basis may cease in the respective circumstances. When providing inspection services by halting the customer's production lines becomes too costly for some reason, for instance, the product design may benefit from more modular structures.

The evaluation (Step 9) can be better supported by BDA application (types PP, PS, SP, and SS). For example, the BDA of the technical performance (e.g., product downtime, functional efficiency) or environmental performance (e.g., carbon emission) of a previous or current generation product in use (PP and PS) or customer feedback regarding a previous or current generation service (SP and SS) can be used as a baseline to evaluate the efficacy of the PSS design concepts for a specific circumstance in question. BDA use is even more relevant for evaluating a fleet of products provided as a PSS [88].

The other steps (Step 5, 8, and 10) are not deemed highly relevant because designers in these steps are supposed to mostly use the outputs from the previous steps. For instance, Step 8 (selecting combinations) is executed on the given alternatives, i.e., the output from Steps 6 and 7.

*4.2. Systematic Review and Inter-Disciplinary Meta-Synthesis of BDA Literature with PSS Design Schema*

4.2.1. Overview of the Results of the Literature Search

From the keyword search defined in Section 3.3, 122 objects were hit, as shown in the Supplementary Material Table S1. Out of these, 117 articles were identified in the set of articles relevant to BDA and design (in a broader sense); see the accessed journals in Appendix B. A major characteristic of the 117 articles was the years of publication: those published in or after 2018 dominated this set with a 74% share, and the mode was the year 2019. These indicate the current growth in terms of the number of publications with this topic and the timeliness of reviewing the work for analyzing future research opportunities. Another characteristic was the variety of disciplines of the journals, spanning from computer science, operations research, engineering, and management, to sustainability. The rest of Section 4.2.1 overviews the literature in this set, focusing on research based on application to industrial cases but leaving the research more relevant to design to Section 4.2.2.

Generally, the literature informs on the positive effects of BDA, while little on the investment, risks, and other potential negative effects is documented. For instance, BDA was found to be a relevant determinant for becoming a product innovator and for the market success of product innovations using German firm-level data [89]. In addition, BDA management capabilities have a strong and significant effect on innovative green product design and sustainable supply chain outcomes based on a survey with companies in South Africa [90].

Production or factory lines were a frequent target of BDA application. A data-driven predictive planning system for production was proposed to demonstrate its superiority in terms of prediction of energy consumption and machining time based on data from physical and virtual shop floors [91]. In addition, for estimating the costs of additive manufacturing, a framework based on BDA was demonstrated with errors between 0.93% and 6.51% [92]. Further, a framework based on Bayesian inference and Gibbs sampling, using the data of 20 lots of 500 wafers in a semiconductor manufacturing line with 100 process stages, revealed potential critical factors influencing the yield in an improved manner than domain knowledge alone used [40].

Supply chain management (SCM) is another target with frequent application of BDA. For instance, weekly sales forecasting was shown to be more precise by applying a predictive model empowered by sentiment analysis on big data of consumer reviews on social media [41]. In addition, a process-focused view, not only attention to big data as such but also how big data are handled in relation to the business processes across actors involved in SCM, was recommended to be used for improving SCM by BDA in practice after the investigation of an SCM case for bananas [93].

4.2.2. Descriptive Results of Design in Industry Enhanced by BDA with Industrial Cases

This subsection describes the research that was based on application to industrial cases and has shown potential in contribution to design in industry. Out of the 117 articles, 11 articles were found to be such research and are summarized in Table 5. The type column shows the elementary type of applying BDA in PSS design, as explained in Table 3, e.g., "source the data of products in use to design service". The research in this table is ordered according to the type, from PP, PS, SS, to SP. The next two columns, "data used" and "aim", explain the type in the context of the specific research. The "method used" and "result" explain the application. The aspects mean the aspects (of outputs or processes of design) with the contribution of the method applied to the industrial case, that is, "quality" (Q or q), "economic cost" (C or c), "lead time" (T or t), and "environmental impacts" (E or e): capitalized letters (Q, C, T, and E) in Table 5 denote explicit indication, while non-capitalized ones are for implicit indication.

**Table 5.** Research that applied BDA to industrial cases and is potentially useful to design in industry.

| Ref. | Type | Data used | Aim | Method Used | Result | Aspects [1] | | | |
|---|---|---|---|---|---|---|---|---|---|
| [94] | PP | Operating conditions and the responses (e.g., a steady-state condition for 120 minutes with 5-minute interval) | Optimize operations in oil re-refining processes | Principal component analysis | Fewer experiment sets for design of experiment: eventual improvements were quantified in terms of product yield (e.g., 84% increased), process quality (e.g., 47% increased), environmental impacts (e.g., 91% improved in acidification potential), etc. | Q | C | T | E |
| [95] | PP | 26,000 customers involved: Social media data (130,000 comments) and machine-generated/sensor data | Better understand customers in NPD at a wearable medical equipment manufacturer | Data (including text) mining and clustering analysis | More precise insights of customer perceptions (trends, expectations, preferences, etc.), leading to NPD with <5 months, that is, less than half in time and 2 million USD (a fraction of economic cost) compared to a traditional NPD project | Q | C | T | |
| [96] | PP | Seven peer brands' sugar levels and prices, and consumer feedbacks on lemonade quality | Support NPD for a lemonade with a reduced sugar level | Text mining | Reducing the NPD costs by 33% and time by 10% without compromising the quality | Q | C | T | e |
| [97] | PP | Customers' preferences registered via the official company website, product information (including videos), social media (10 million threads), and customer locations. | Better manage competences for NPD in a Chinese manufacturer of athletics goods owning three main manufacturing facilities. | Data mining | Strategies such as an optimal expansion on the company's existing competences by considering internal and external competences: seven golf clubs (e.g., 3-iron and 460cc 10.5° driver) were analyzed with 14 competences (e.g., wood manufacturing and stamping technologies). | Q | C | T | |
| [98] | PP | 100,000 user evaluations of fitness mobile apps on social media | Analyze competing products | Natural language processing and machine learning | Four clusters of ca 500 apps and an app's position in relation to peers, including the product functionality similarity, contributing to making strategies on pricing models, product differentiation, and faster adaptation | Q | C | T | |
| [99] | PP PS | Data of one million bridges in the USA; e.g., structural types, condition ratings, geographical zones, and traffic volumes | Analyze and evaluate conditions (incl. degradation) of constructed bridges | ANOVA (analysis of variance) | Insights such as the adequate selection of structural types (e.g., concrete cast-in-place) dependent on the use environments, enhancing the performance and longevity of the bridges, and planning better inspection/maintenance based on deterioration | Q | | | e |
| [100] | SS | Purchase transactions by 110,000 customers at this agency and its competitors' pricing data | Analyze customer behaviors and predict their next purchases of flight tickets at an online travel agency | Data mining and customer segmentation | Patterns and correlations in customer purchasing behaviors, contributing to better customer relationship management, such as targeted promotion | Q | c | t | |
| [101] | SS | Data about the software, such as purchase, 330,000 renewals, 120,000 non-renewals, download, problems, and evolution | Predict the risks of software license cancellation in combination with domain knowledge | Machine learning | The framework for prediction tested at IBM, attesting the usefulness of the framework in industry | q | c | t | |
| [102] | SS | Three million tweets on Twitter by 1200 companies and their financial data | Examine the relationships of product-related communication and the financial performance of manufacturing firms | Text mining | Positive association between divulging product-related information and the firm value on the market | | c | | |
| [103] | SS | 4900 online reviews on TripAdvisor for 200 Spa hotels | Segment hotel customers and the prediction of their choices | Machine learning and clustering | Accurate prediction of user choices per segment, which is expected to contribute to optimal marketing expenditures | Q | c | T | |
| [104] | SS SP | 2600 online reviews on TripAdvisor for 20 hotels in Taipei | Understand hotel guests' perceptions | Text mining | Through extracting Kansei words (e.g., excellent and friendly) and hotel service characteristics (e.g., facilities and service delivery) as well as their relationships, a guideline for hotel service development was proposed | Q | C | T | |

[1] Capitalized letters (Q, C, T, and E) denote explicit indication of the aspects in each publication with or without quantification, while non-capitalized letters are for implicit indication: Q or q for quality, C or c for economic cost, T or t for lead time, and E or e for environmental impacts. NPD stands for new product development. The acronyms for the types are explained in Table 3.

As shown in Table 4, the frequent types were PP (sourcing the data of products in use to design product) and SS (sourcing the data in production and use of services to design service). These two types are BDA application within the product/service boundary. The types of applying BDA across the product/service border were found with only two research works, PS (sourcing the data of products in use to design service) and SP (sourcing the data in production and use of services to design product). This contrast indicates the higher potential of PSS design thinking (across the product/service border) in BDA application: that with PS addressed the better design of inspection/maintenance service based on product deterioration, while that with SP developed a guide to better design facility (i.e., products) in a hotel based on the feedback to services. Other observations included that the majority of the data used were high volume data, and less research addressed the other Vs, including velocity and variety. Also, the major methods used were data mining (6 out of the 11 works, of which 4 addressed text) and machine learning (3 out of 11). Most of the research addressed positive aspects in quality, cost, and lead time, either directly or indirectly. The environmental aspect was addressed by only three research works, showing a higher potential for research.

4.2.3. Discovering Research Opportunities in PSS Conceptual Design Enhanced by BDA

The theoretically derived relevance of the PSS design steps to the BDA application shown in Table 4 are reflected in the industrial applications of BDA shown in Table 5. Namely, BDA theoretically applicable to the PSS design schema is compared with the BDA applications reported in the literature thus far. Table 6 classifies the 10 steps into those where the BDA applications reported in the literature are considered useful and those with no industrial application reported thus far: the latter means future research opportunities (indicated by "O"s) discovered for PSS design and BDA as an outcome of the inter-disciplinary literature meta-synthesis.

**Table 6.** Correspondences of the earlier BDA applications to the PSS conceptual design schema.

|  | PP (Product to Product) and PS (Product to Service) | SP (Service to Product) and SS (Service to Service) |
|---|---|---|
| Step 1: Functional unit definition | O | O |
| Step 2: Stakeholder identification |  | [103] |
| Step 3: Requirement consolidation | [95,96] | [100,104] |
| Step 4: Value proposition | [96,98] | [100] |
| Step 5: Criterion identification |  |  |
| Step 6: Element integration | [99] [1] | O |
| Step 7: Balance examination | O | O |
| Step 8: Selecting combinations |  |  |
| Step 9: Evaluating combinations | O | O |
| Step 10: Solution selection |  |  |

[1] Ref. [99] corresponds to conceptualization as such and not integration on the PSS context (see Section 4.1.2). Note: O means a research opportunity.

For instance, reference [95] showed the possibility of using BDA to obtain useful insights into customers, such as customer expectations and preferences on products (see details in Table 5), and this research shows that BDA is useful in Step 3 (requirement consolidation). Reference [98] documented the use of BDA for making differentiation strategies of a product being designed based on the comparison of product features within a derived cluster of products, which shows the usefulness of BDA in Step 4 (value proposition). Reference [103] reported BDA used for identifying similar consumers, and this shows BDA is useful for Step 2 (stakeholder identification). For the other steps from those marked in Table 4 (i.e., Steps 1, 7 and 9), no application is reported in the scientific literature thus far, and, therefore, research opportunities are discovered. Note that four works in Table 5 are not shown in Table 6, as they were deemed not so relevant to the conceptual design of PSS: reference [94] improved the detailed design of production processes, reference [101] addressed sales operations, and references [97,102] addressed mainly firm-level activities.

## 5. Discussion

### 5.1. Scientific Contributions

This article proposed a prescriptive schema as a consolidated procedural support for conceptual PSS designing. Sections 4.1.1 and 4.1.2 gathered the insights of extant prescriptive PSS design knowledge through a meta-analysis and subsequent intra-disciplinary synthesis of existing reviews of the state of the art. The insights gathered from this synthesis were consolidated into the mold of a systematic design approach (PBSA) to form a schema with 10 steps that are recommended to be followed methodically yet flexibly during the conceptual design phase. This type of representation of consolidated and granular knowledge was found to be missing in the extant literature and was deemed necessary for conceptual PSS design in an earlier review by Vasantha, Roy, and Corney [34]. While in its current format this schema can be utilized as a conceptual PSS design domain-specific support for practitioners who are not experienced in PSS design, it also creates opportunities to initiate debates for possible amendment or adjustment and thereby can potentially contribute to advance the knowledge of PSS design.

In addition, we initially identified the potential for enhancing each of the 10 steps of the PSS design schema with BDA, using qualitative analysis in Section 4.1.3. As shown in Table 4, Steps 1, 2, 3, 4, 6, 7, and 9 were identified qualitatively by the authors to have the potential to be enhanced by BDA, while the other steps were evaluated to be insufficiently relevant. This qualitative analysis was followed by the mapping of existing BDA research with industrial case applications (in Table 5), in relation to the four BDA application types for PSS design (identified in Table 4). This mapping of extant applied BDA research identified practical and concrete ways in which BDA has been previously exploited to enhance certain aspects of the PSS design process. This mapping revealed how various types of incoming data and information from the use phases of products and services (potentially) supported the design of products or services, or both, in industrial contexts. This outcome has a higher granularity than the results reported in the extant literature. That is, most earlier studies of relevance [2,105,106] were unspecific with regards to the application of BDA to the specific procedural aspects of PSS design; e.g., some of these previous studies reported that BDA supports the reduction of time to market for the whole design, and seldom indicated which concrete steps of conceptual design can potentially benefit from the obtained data and information. Thereby, RQ1, concerning the relationship between PSS design and BDA, was answered. This advancement partly fills the knowledge gap for design using smartness [107], i.e., the insufficient connection between service design and product design as PSS design.

Further, Table 6 indicates that Steps 2, 3, 4, and 6 of the PSS design schema benefit from BDA, as reported in the previous research with industrial cases. Even though Steps 1, 6, 7, and 9 were identified to have the potential to utilize BDA in Table 4, the results of the systematic literature review and subsequent analysis indicate that there is a lack of research reporting the application of BDA for these steps, as shown in Table 6. This lack of research points towards opportunities for future applied research to support these specific steps of the PSS design schema. Also, the lack of research reporting the application of BDA to support the integration of product and service elements, which is a peculiar characteristic of PSS design, should be emphasized. These past reports of application, and the identified potential opportunities for BDA application to the different steps of the PSS design schema, answer RQ2. This answer reflects on past achievements and future development opportunities for BDA application in PSS design.

The reflections described above indicate that the PSS design schema was, through the application to BDA as an example, shown to be effective. Without such a generic description, it would have been impossible to connect PSS design and BDA applications in a generic manner. This also indicates that this consolidated design schema can be used to direct further exploitation of BDA in PSS design without developing a new PSS design process dedicated to using BDA. An investigation of the impacts from this consolidation of generic insights of existing prescriptive PSS design research on the cognitive

nature of conceptual PSS designing was already begun by the authors' group [108] and will be scientific future research of relevance.

This article showed a relatively new method concerning research methods using the literature. For the intra-disciplinary literature review, the major novelty lies in its concrete way to perform synthesis, as such. A major strength lies in the practicality of the outcome represented by the 10 steps. Also, the inter-disciplinary literature review, as such, is relatively novel. In particular, the inter-disciplinary literature synthesis that was enabled by the practical form of the outcome from a discipline (as stated in the previous paragraph) is novel. The novelty will contribute to advancing the knowledge of literature-based methods, especially seeing a high potential to improve the usefulness of review articles (as explained in Section 2.3). Clearly, a more specifically pre-defined procedure could be applied for the intra-disciplinary literature synthesis of PSS design, for instance, by documenting the themes and topics addressed by each original literature cited in a review. However, the meaning of the literature synthesis lies in a high-level consolidation, and therefore the level of transparency shown was deemed sufficient for the purpose of this article.

### 5.2. Environmental Implications

In answering RQ3, concerning the environmental implications of BDA application in PSS design, this article found that three previous research works with industrial cases reported positive contributions, either explicitly [94] or implicitly [96,99]. This article advanced the compilation of research with regards to this question, as other reviews such as [16] did not explicitly identify and report on such previous research effort. It may be fair to state that this type of research is underachieved, especially considering the high needs from the societies on using intelligent technologies to improve the environmental sustainability performance of industrial activities (e.g., [109]).

### 5.3. Implications of BDA on Customization

An earlier review showed what and how to design in the context of PSS customization [110]. In addition, the general effectiveness of BDA for PSS customization was acknowledged, as pointed out in Section 2.2. Building upon these insights, this article provided specificity of the effectiveness of BDA on PSS customization in terms of conceptual PSS design steps. Especially, Step 2 (stakeholder identification), Step 3 (requirement identification), and Step 4 (value proposition) are highly relevant to customization, as shown by the customization literature: Step 2 can include segmentation of potential customers [111], and carrying out Steps 3 and 4 with differentiation between the segments form a central process of customization, as modeled in [112]. In this way, this article showed how to connect the customization literature with BDA. Knowledge of customization with the environmental aspects, such as [113,114], can be further exploited using BDA.

### 5.4. Practical Contributions

The practical contributions of this article are threefold. First, the presented 10-step schema for conceptual PSS design incorporates essential elements from PSS design acknowledged in academia. It is not exhaustive in relation to each publication of relevance. The schema, however, represents the essence in a condensed form and, thereby, provides a practitioner with an opportunity for efficient comprehension of past achievement in PSS design. Therefore, academic knowledge exploitation can be more efficient in industrial practice. This type of exploitation is carried out, for instance, when practitioners aim to improve an existing design process or R&D (research and development) process of a company in question (see such an instance in [15]). Second, the insight of this article helps practitioners with one way to adopt BDA and capture value in a well-planned manner with a systems perspective. This is enabled through employing PSS design, which is a kind of planning of long-term (e.g., a product lifecycle) by nature, and builds upon a systems perspective. Thereby, these insights ensure that practitioners would avoid sub-optimization in a certain phase of the lifecycle or by one single function (or department) of a company. Third, the reasoning above can be applied beyond

the level of individual practitioners: based on the same reasoning, this article could be useful for a public organization to develop an industry standard for PSS design or BDA application. In particular, concerning PSS design, where only national standard documents are available, such as PAS 2009 [115], despite its maturity in industry practice, a potential to create an international standard is observed.

## 6. Conclusions

A PSS conceptual design schema using literature synthesis was proposed as a representation of consolidating a variety of PSS design support. A 10-step procedure was adopted to represent the existing insights of PSS conceptual design in a condensed form. This schema, as well as the literature synthesis, was shown effective to increase the usability and accessibility of the knowledge, through application to the analysis of BDA literature. One challenge in creating the schema to be noted was the balance needed to be taken between the comprehensiveness of the PSS design insights and the practicality of the PSS design schema. In addition, through the systematic literature review for BDA, applied research with BDA was connected with the PSS design schema. Thereby, this article gave one perspective between PSS design and BDA, as well as presented an account of an interdisciplinary, high-level review for PSS design enhanced by BDA. Theoretically possible applications of BDA onto the PSS design schema were discovered: some of the possible applications were found in extant BDA literature, while others exhibited potential for future work. This way, future related research could be anchored, motivated, and planned with more specificity. In particular, the environmental aspect provides a high potential to be researched. One specific future research of interest can be analysis of the contribution of each step of the PSS design schema in terms of quality, cost, delivery (time), and sustainability, since Table 5; Table 6 touch upon this issue. As a limitation, it can be pointed out that the relationship between PSS design and BDA presented in this article is one of the various possible relationships. Readers are invited to investigate other possible relationships. Lastly, practical challenges for applying BDA in PSS conceptual design are foreseen concerning ownership of the big data. For instance, do PSS designers have access to the relevant big data? If not, what mechanisms could solve this accessibility issue? This might be also an interesting scientific research topic.

**Supplementary Materials:** The following are available online at http://www.mdpi.com/2071-1050/12/8/3484/s1; Table S1: BDA_literature.xlsx.

**Author Contributions:** The first conceptualization and methodology, T.S.; writing—original draft preparation, A.N. mainly for the PSS design and T.S. mainly for the BDA, while the intersection was addressed jointly by A.N. and T.S.; review performed by both A.N. and T.S.; funding acquisition, T.S. All authors have read and agreed to the published version of the manuscript.

**Funding:** This research was supported in part by the Mistra REES (Resource-Efficient and Effective Solutions) program, funded by Mistra (The Swedish Foundation for Strategic Environmental Research) (grant number DIA 2014/16).

**Acknowledgments:** An earlier and shorter version of this paper was presented at the 8th Spring Servitization Conference in Linköping in May, 2019 with the title "Enhancing PSS Design through Big Data, IoT, and Big Data Analytics". Yang Liu of Linköping University contributed to data collection.

**Conflicts of Interest:** The authors declare no conflict of interest.

## A List of Acronyms

| | |
|---|---|
| BDA | big data analytics |
| ISI | Institute for Scientific Information |
| MTBF | mean time between failures |
| MTTF | mean time to failure |
| NPD | new product development |
| PBSA | Pahl's and Beitz' systematic approach |
| PSS | product/service system |
| R&D | research and development |
| RQ | research question |

SCI            Science Citation Index
SCM           supply chain management
SSCI          Social Sciences Citation Index

## Appendix A

Refined by: [excluding] WEB OF SCIENCE CATEGORIES: (CARDIAC CARDIOVASCULAR SYSTEMS OR POLYMER SCIENCE OR CELL BIOLOGY OR VETERINARY SCIENCES OR CHEMISTRY ANALYTICAL OR BIOCHEMISTRY MOLECULAR BIOLOGY OR CHEMISTRY APPLIED OR CHEMISTRY MEDICINAL OR IMMUNOLOGY OR CRIMINOLOGY PENOLOGY OR INSTRUMENTS INSTRUMENTATION OR DERMATOLOGY OR MEDICINE RESEARCH EXPERIMENTAL OR NUTRITION DIETETICS OR OPTICS OR FAMILY STUDIES OR CHEMISTRY MULTIDISCIPLINARY OR PEDIATRICS OR GASTROENTEROLOGY HEPATOLOGY OR NANOSCIENCE NANOTECHNOLOGY OR REHABILITATION OR GENETICS HEREDITY OR ENERGY FUELS OR SURGERY OR GEOGRAPHY PHYSICAL OR BEHAVIORAL SCIENCES OR GEOSCIENCES MULTIDISCIPLINARY OR PHYSICS APPLIED OR COMPUTER SCIENCE CYBERNETICS OR GERIATRICS GERONTOLOGY OR HEMATOLOGY OR ENDOCRINOLOGY METABOLISM OR INFECTIOUS DISEASES OR ENGINEERING BIOMEDICAL OR INFORMATION SCIENCE LIBRARY SCIENCE OR PHARMACOLOGY PHARMACY OR HEALTH POLICY SERVICES OR INTERNATIONAL RELATIONS OR RHEUMATOLOGY OR MEDICAL INFORMATICS OR MATERIALS SCIENCE BIOMATERIALS OR CHEMISTRY PHYSICAL OR NEUROSCIENCES OR MATERIALS SCIENCE CHARACTERIZATION TESTING OR ONCOLOGY OR MATERIALS SCIENCE TEXTILES OR ORTHOPEDICS OR PERIPHERAL VASCULAR DISEASE OR MATHEMATICAL COMPUTATIONAL BIOLOGY OR CLINICAL NEUROLOGY OR PHYSICS CONDENSED MATTER OR MICROBIOLOGY OR MEDICINE GENERAL INTERNAL OR REMOTE SENSING OR MINING MINERAL PROCESSING OR HEALTH CARE SCIENCES SERVICES OR SPORT SCIENCES OR NUCLEAR SCIENCE TECHNOLOGY OR AUTOMATION CONTROL SYSTEMS OR UROLOGY NEPHROLOGY OR NURSING OR WATER RESOURCES OR OBSTETRICS GYNECOLOGY OR ENGINEERING CHEMICAL OR AGRONOMY OR OTORHINOLARYNGOLOGY OR OPHTHALMOLOGY OR PLANT SCIENCES OR BIOCHEMICAL RESEARCH METHODS OR PSYCHIATRY OR BIOPHYSICS OR PSYCHOLOGY DEVELOPMENTAL OR FOOD SCIENCE TECHNOLOGY OR BIOTECHNOLOGY APPLIED MICROBIOLOGY OR PUBLIC ENVIRONMENTAL OCCUPATIONAL HEALTH OR INTEGRATIVE COMPLEMENTARY MEDICINE).

## Appendix B

ACM SIGCOMM COMPUTER COMMUNICATION REVIEW; ACM TRANSACTIONS ON INTERNET TECHNOLOGY; ADVANCES IN PRODUCTION ENGINEERING & MANAGEMENT; ANNALS OF OPERATIONS RESEARCH; APPLIED ENERGY; APPLIED SCIENCES-BASEL; APPLIED SOFT COMPUTING; AUTOMATION IN CONSTRUCTION; BRITISH FOOD JOURNAL; BRITISH FOOD JOURNAL; BUSINESS PROCESS MANAGEMENT JOURNAL; CHINA COMMUNICATIONS; CIRP ANNALS-MANUFACTURING TECHNOLOGY; COMPUTERS & INDUSTRIAL ENGINEERING; COMPUTERS IN INDUSTRY; DATA TECHNOLOGIES AND APPLICATIONS; DISTRIBUTED AND PARALLEL DATABASES; ECONOMICS OF INNOVATION AND NEW TECHNOLOGY; ELECTRONICS; ENGINEERING; ENGINEERING STRUCTURES; EUROPEAN JOURNAL OF OPERATIONAL RESEARCH; EXPERT SYSTEMS WITH APPLICATIONS; FRONTIERS IN MEDICINE; FRONTIERS OF MECHANICAL ENGINEERING; HEALTH SECURITY; IEEE ACCESS; IEEE COMMUNICATIONS SURVEYS AND TUTORIALS; IEEE JOURNAL OF BIOMEDICAL AND HEALTH INFORMATICS; IEEE TRANSACTIONS ON CYBERNETICS; IEEE TRANSACTIONS ON INDUSTRIAL INFORMATICS; IEEE TRANSACTIONS ON INTELLIGENT TRANSPORTATION SYSTEMS; IEEE TRANSACTIONS ON VISUALIZATION AND COMPUTER GRAPHICS; INDUSTRIAL MANAGEMENT & DATA SYSTEMS; INDUSTRIAL MARKETING MANAGEMENT; INFORMATION & MANAGEMENT; INFORMATION AND SOFTWARE TECHNOLOGY; INFORMATION SYSTEMS FRONTIERS; INTERNATIONAL JOURNAL OF ADVANCED MANUFACTURING TECHNOLOGY; INTERNATIONAL JOURNAL OF COMMUNICATION SYSTEMS; INTERNATIONAL JOURNAL OF DIGITAL EARTH; INTERNATIONAL JOURNAL OF HOSPITALITY MANAGEMENT; INTERNATIONAL JOURNAL OF INFORMATION MANAGEMENT; INTERNATIONAL JOURNAL OF LOGISTICS MANAGEMENT; INTERNATIONAL JOURNAL OF OPERATIONS & PRODUCTION MANAGEMENT; INTERNATIONAL JOURNAL OF PRODUCTION ECONOMICS; INTERNATIONAL JOURNAL OF PRODUCTION RESEARCH; INTERNATIONAL JOURNAL OF RETAIL & DISTRIBUTION MANAGEMENT; JOURNAL OF AIR TRANSPORT MANAGEMENT; JOURNAL OF AMBIENT INTELLIGENCE AND SMART ENVIRONMENTS; JOURNAL OF BUSINESS & INDUSTRIAL MARKETING; JOURNAL OF BUSINESS RESEARCH; JOURNAL OF CLEANER PRODUCTION; JOURNAL OF COMPUTER INFORMATION SYSTEMS; JOURNAL OF ENTERPRISE INFORMATION MANAGEMENT; JOURNAL OF INFORMATION SCIENCE; JOURNAL OF KNOWLEDGE MANAGEMENT; JOURNAL OF MANUFACTURING SYSTEMS; JOURNAL OF MEDICAL SYSTEMS; JOURNAL OF RESEARCH IN INTERACTIVE MARKETING; JOURNAL OF SERVICE RESEARCH; LIBRARY HI TECH; MANAGEMENT DECISION; MEASUREMENT; PERVASIVE AND MOBILE COMPUTING; PROCESS SAFETY AND ENVIRONMENTAL PROTECTION; PRODUCTION AND OPERATIONS MANAGEMENT; QUALITY AND RELIABILITY ENGINEERING INTERNATIONAL; RAPID PROTOTYPING JOURNAL; RENEWABLE & SUSTAINABLE ENERGY REVIEWS; RESOURCES CONSERVATION AND RECYCLING; ROBOTICS AND COMPUTER-INTEGRATED MANUFACTURING;

SENSORS; SUSTAINABILITY; SUSTAINABLE CITIES AND SOCIETY; TRANSPORT POLICY; WILEY INTERDISCIPLINARY REVIEWS-DATA MINING AND KNOWLEDGE DISCOVERY.

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
