# Peer review of "A Product/Service System Design Schema: Application to Big Data Analytics"

_sustainability, doi:10.3390/su12083484_

Round 1

Reviewer 1 Report

Dear Authors

Thank You for this very interesting paper. It gives a good overview of PSS and of the possibilities of Big Data to improve this process and come up with more relevant PSS propostions. 

Though... I have some recommendations. 

A general recommentation is that 'sustainability' and environmental issues are only adressed at the side line. This could be improved or suggestions for a more inherent sustainable approach could be formulated. 

On page 6-246 you mention the importance or relevance of a business model. Later I don't really find this back in the 10 steps? Can you incorporate this somehow?

This together with the previous remark could open up opportunities to integrate a more overall environmental friendly reflection. 

Another general remark is the place (step 4) you give to 'value' in the process of PSS development. Isn't this due to the consultation of literature that was more focussed on PP development that you come up to first take 3 other steps? When developing services, isn't it more inherent to first reflect on the added value before reflecting on the functionality? Isn't that (among others) what the product-service continium is about or the product-service layers are about.

In section 3.2. and the sections following from this description, I suggest to ad a table (figure) with a nice literature overview and the most important outcomes. 

Now you build up a 7.. and later a 10 step model that seems rather obvious and well known. Explain better why these steps are important as this is the core of your paper and refer hereby  to literature, examples .. in PSS context. Make a table or a figure with the steps, their key words, their main references, relevance for PSS/PP/ Reflect if there is a contribution to sustainability, to time to market, to market succes .. or other KPI's in dialogue with this recommendations from literature. What parameters, variables, outcomes.. are relevant? 

In Table 2, what do you mean with T? 

In Table 3, what do you mean with QCTE? 

At page 10, you give an overview of the domains BD is applied in de design process. Analyse this further in relation to the 10-step plan. 

As this is a literature study, I suggest to give more attention to further research. How can your research feed research questions, what are the gaps, what methods seemed promissing, which didn't, etc... 

You should provide a hands-on and clear pathway for other researchers to continue in this research field. 

This part should be elaborated more in depth. 

But apart from this recommendations, I liked the paper and the efforts that are made and I think it is a good angle to look at the integration of BD! 

Thank you

Reviewer 2 Report

The paper deals with the optimisation of the conceptual design process applied product/service systems in connection with the big-data analytics (BDA) approach. The paper main result is a novel 10 steps conceptual design scheme that considers the role and benefits of BDA integration into the conceptual design process.

The paper has the following strengths:

  1. The problem is well and clearly justified and formulated
  2. The approach is clearly explained
  3. The paper results are extensively detailed and interpreted
  4. The work as a whole is very clear and brings important contributions in the domain

The following issues are recommended to improve the paper:

  1. Line 83-84: recommendation to define “conventional product design” and the major difference in relation with “concurrent/integrated product design”
  2. It’s not really clear your connection of environmental sustainability (as paper keyword) with PSS design topic. The environmental sustainability is introduces only as a driver of PSS or other roles are intended here?
  3. Line 346: define the acronym “MTBF”
  4. Define T in Table 2.
  5. The acronyms for the Aspects in Table 3 are not yet defined in the paper (maybe better bellow the table).

Reviewer 3 Report

  1. The authors should clearly highlight the added value to the literature by the implementation of the review. Which are the main contributions and the novel characteristics of the review conducted to the scientific community?
  2. Are there any challenges that need to be addressed in the future? Please highlight them.
  3. The conclusions section should be totally redesigned. At its current form, it contains a synopsis of the work conducted. The authors should clearly cite the key findings of the work, and the challenges they met during its implementation.
  4. An addition of an Acronyms section is highly recommended to facilitate the reader.
  5. Could you please elaborate more on the usefulness of the conducted work for industries, utilities, and policymakers?
  6. Could you provide some additional information regarding the best global practices currently applied?

Reviewer 4 Report

This review gives a PSS design schema and a perspective nexus between PSS design and BDA by using systematic literature review and literature synthesis method. Overall, this manuscript has reviewed sufficient references and in-depth analysis of PSS and big data technologies. This paper has a certain scientific significance in related fields. I recommend this paper to be published in Sustainability.

Just one comment: Can the authors provide more prospects for research in PSS and BDA?
